# The Potential Role of m^6^A in the Regulation of TBI-Induced BGA Dysfunction

**DOI:** 10.3390/antiox11081521

**Published:** 2022-08-04

**Authors:** Peizan Huang, Min Liu, Jing Zhang, Xiang Zhong, Chunlong Zhong

**Affiliations:** 1Department of Neurosurgery, Shanghai East Hospital, School of Medicine, Tongji University, Shanghai 200120, China; 2Department of Neurosurgery, The Fourth Affiliated Hospital of Nanjing Medical University, Nanjing 210031, China; 3Institute for Advanced Study, Tongji University, Shanghai 200092, China; 4College of Animal Science and Technology, Nanjing Agricultural University, Nanjing 210095, China

**Keywords:** m^6^A RNA modification, brain-gut axis, traumatic brain injury

## Abstract

The brain–gut axis (BGA) is an important bidirectional communication pathway for the development, progress and interaction of many diseases between the brain and gut, but the mechanisms remain unclear, especially the post-transcriptional regulation of BGA after traumatic brain injury (TBI). RNA methylation is one of the most important modifications in post-transcriptional regulation. N6-methyladenosine (m^6^A), as the most abundant post-transcriptional modification of mRNA in eukaryotes, has recently been identified and characterized in both the brain and gut. The purpose of this review is to describe the pathophysiological changes in BGA after TBI, and then investigate the post-transcriptional bidirectional regulation mechanisms of TBI-induced BGA dysfunction. Here, we mainly focus on the characteristics of m^6^A RNA methylation in the post-TBI BGA, highlight the possible regulatory mechanisms of m^6^A modification in TBI-induced BGA dysfunction, and finally discuss the outcome of considering m^6^A as a therapeutic target to improve the recovery of the brain and gut dysfunction caused by TBI.

## 1. Introduction

Traumatic brain injury (TBI), one of the major causes of death and disability worldwide [1], impacts more than 50 million people every year, of whom 10 million die and nearly another 30 million remain physically disabled. Survivors experience a huge influence of physical, mental and cognitive disabilities, bringing considerable costs to their families and society [2,3]. In China, over 1 million new cases of TBI are reported each year [4]. In recent years, although the number of deaths caused by TBI has decreased significantly, unfortunately, effective treatments to promote the restoration of brain functions are still lacking. Thus, there is currently a major shift in TBI research in neural recovery and neurorehabilitation [5]. Increasing evidence has confirmed that the brain–gut axis (BGA) plays a key role in regulating behaviours associated with various mental and neurological diseases [6,7,8], including TBI [9]. A meta-analysis revealed that downregulated levels of anti-inflammatory butyrate-producing bacteria and upregulated levels of pro-inflammatory genera were consistently shared between major depressive disorder, psychosis and schizophrenia, bipolar disorder, and anxiety [10]; while a study showed that rifaximin improved depressive-like behaviour induced by chronic unpredictable mild stress via regulating the abundance of faecal microbial metabolites and microglial functions [11]. A clinical study showed significant alterations in gut microbiota and metabolome in faecal samples from Parkinson’s disease(PD) patients [12], while a meta-analysis indicated that inflammatory bowel disease(IBD) may increase PD risk, especially in patients over 65 years of age [13]. A pilot study suggested that the diversity of the microbial ecosystem in brain tumour patients was less than the healthy controls [14], while caecal content transfer experiments in mice revealed that cancer-associated alteration in caecal metabolites was involved in late-stage glioma progression [15]. In addition, there is clear evidence that TBI can cause gut microbiota dysbiosis [16], while disruption of the gut barrier and depletion of the gut microbiota are associated with neurologic outcomes following TBI [17]. With a growing understanding of BGA, people have a strong interest in studying the influences of BGA in neural recovery and neurorehabilitation after TBI.

However, the mechanisms of BGA in regulating mental and nervous system diseases induced by TBI are still unclear, especially in posttranscriptional regulation. Methylation modification is one of the most essential modifications in posttranscriptional regulation, including N6-methyladenosine (m^6^A), N1-methyladenosine, inosine, 5-methylcytidine, 5-hydroxymethylcytidine, pseudouridine, N6,2′-O-dimethyladenosine, N4-acetylcytidine, N7-methylguanosine, 8-oxoguanosine and 2′-O-methyl [18,19,20]. Among them, m^6^A, one of the most abundant mRNA posttranscriptional modifications in eukaryotes, has recently been identified and characterised in both the brain and gut. M^6^A modification is prevalent in the central nervous system, regulates the activation of various nerve conduction pathways and plays an irreplaceable role in the development, differentiation and regeneration of neurons [21,22,23]. At the same time, it exhibits an important impact on the communication between the gut microbiome and the host [24,25,26,27]. Therefore, m^6^A modification may also play a significant role in BGA. The main aim of this paper is to summarise the roles of m^6^A RNA methylation in post-TBI BGA to further highlight the possible regulatory mechanisms of m^6^A modification in TBI-induced BGA dysfunction, and, finally, to discuss the outcome of considering m^6^A as a therapeutic target to improve the recovery of the brain and gut dysfunction caused by TBI.

## 2. BGA

BGA, a bidirectional communication network connecting the central nervous system (CNS) and enteric nervous system (ENS) [28], is mainly composed of three pathways: the neural, neuroendocrine and immune pathways (Figure 1) [29]. (1). The neural pathway: the bidirectional exchange between the CNS and ENS, is the basis of BGA, and the autonomic nervous system (ANS), whose main branch is the ENS, is the centre of this interaction [9]. The afferent nerves of these systems are in key locations throughout the gut, including the mucosal surface. Both the sympathetic and parasympathetic systems are active, and the imbalance of input induces autonomic nerve disorders, which leads to gut dysfunction [30]. In addition, the gut microbiome can affect the production of neurotransmitters by “de novo synthesis” or by impacting neurotransmitter-related metabolism pathways [29]. (2). The neuroendocrine pathway: the hypothalamus-pituitary-adrenal (HPA) axis, is the key to neuroendocrine transmission and the stress response system [31,32]. Both the neural and nonneural components of the HPA axis have multiple regulatory controls to ensure that the response is suitable for both stressful and non-stressful stimuli, such as appetite, sexual experience, and even light exposure [33]. Meanwhile, the gut microbiome is indispensable in developing and functioning the HPA axis. The gut microbiome directly affects the generation of glucocorticoids and immune mediators, such as tumour necrosis factor-alpha (TNF-α) and interleukin-1 beta and 6 (IL-1β and IL-6), and then activates the HPA axis [34]. (3). Immune pathway: highly specialised macrophages in the CNS constitute microglia that release proinflammatory cytokines, such as TNF-α, IL-1 and IL-6, to maintain microglia-mediated inflammation and recruit other immune cells [35] in response to stress, thereby leading to gut maladaptation and inflammation. Furthermore, the development and integrity of the blood–brain barrier (BBB) and gut barrier depend on the gut microbiome. Changes in the gut microbiome downregulate the expression of tight junctions (TJs) [36], disclosing both organs to microorganisms and biomacromolecules and inducing neuroinflammatory processes [37].

The bidirectional effects of BGA have been studied for an increasing number of neuropsychiatric diseases, including PD [38,39,40], Alzheimer’s disease (AD) [41,42], amyotrophic lateral sclerosis (ALS) [43,44,45] and multiple sclerosis (MS) [46,47,48]. (1). PD: compared to controls, the faecal aromatic amino acids and branched chain amino acids (BCAAs) concentrations are significantly reduced in PD patients [12]. Substantial lines of evidence have confirmed that inducible nitric oxide synthase (iNOS) is produced when lipopolysaccharides (LPS) are increased in substantia nigra [49]. In addition, a damaged gut barrier enhances levels of IL-1β, IL-6 and TNF-α in PD [50]. Gut dysbiosis induces microglial activation, which leads to the subsequent progression of PD [39,40]. (2). AD: a meta-analysis showed a significant reduction in gut microbiota diversity in patients with AD compared to healthy controls [51]. During ageing, the bacteria inhabiting in the gut releases mountains of LPS and amyloids when both the gastrointestinal tract and the BBB are more permeable [50]. Meanwhile, enhanced gut permeability upregulates the levels of proinflammatory cytokines, such as IL-1β, IL-6, IL-8, IL-12A, IL-18 and TNF-α, in the brain [52]. A study indicated that Bacteroides fragilis lipopolysaccharides (BFLPS) potentially activates the pro-inflammatory transcription factor nuclear factor-kappaB (NFκB), which leads to inflammation-mediated neurodegeneration [53]. (3). ALS: a clinical study revealed that compared to their spouses, the gut microbial communities of ALS patients are more diverse and are deficient in *Prevotella* spp. [54]. Additionally, due to augmented membrane permeability, increased invasion of macrophages and mast cells, and upregulated levels of cyclooxygenase-2 (COX2) result in inflammation in the brain [55]. (4). MS: a main pathological characteristic of MS is gut microbiota dysbiosis [47,48], while the alteration in the gut microbiome reduces C-C chemokine receptor type 9^+^ (CCR9^+^) cells, which express transcription factor c-Maf in high levels, leading to an increased expression of retinoic acid-related orphan receptor gamma t (RORγt) and proinflammatory cytokine IL-17/interferon-γ (IFNγ) in secondary progression MS (SPMS) patients [46] (Figure 2). Indeed, the impacts of BGA on the above diseases may be related to metabolic regulation and could trigger the release of many inflammatory transcription factors to activate the immune pathway, causing neurodegeneration [39,44,46,52,56]. However, the relationship between BGA and m^6^A RNA modification in these diseases is unknown.

## 3. TBI and BGA

TBI-related pathophysiological influences directly associated with gut dysfunction have been increasingly studied. These influences significantly affect the host because BGA is considered to ensure the main bidirectional communication pathway between the brain and gut, including afferent and efferent signals, involving the crosstalk of neurons, hormones and immunity [57,58]. Such bidirectional interactions may cause sequelae, such as chronic fatigue of the gastrointestinal system, including its inability to function properly [59,60] (Figure 3). When TBI impacts this axis, disorders in digestion and absorption, afferent nerve signals, immunomodulation, gut barrier function, and the BBB may occur [5,61]. The effects of TBI on BGA may be realised via the influence of the neural pathway (autonomic nerve), the neuroendocrine pathway and the immune pathway. First, post-TBI autonomic dysfunction causes the altered production and release of key neurotransmitters involved in the sympathetic pathways [62]. When the integration of sympathetic and parasympathetic nerves is disrupted, the gut may also be influenced, involving symptoms of chronic pain and gut discomfort [61] and the dysbacteriosis of normal gut microbiomes. Second, overactivation of the post-TBI HPA axis affects neuroinflammation through the stress-induced initiation of microglia [63]. Dysfunction of the HPA axis occurs immediately after TBI, and the HPA axis is acutely activated by the stress of injury. Excessive glucocorticoids(GCs) caused by extreme activation of the HPA axis after TBI can induce microglial initiation in the brain and increase inflammatory cytokines in response to stress, leading to gut maladaptation and inflammation [63]. TBI activates systemic stress and motivates the HPA axis and the sympathetic branch of the ANS, which results in the release of GCs and catecholamines, respectively [64,65]. Excessive catecholamine release presumably causes gastrointestinal dysmotility involving gastroparesis and food intolerance [9]. Finally, at the cellular level, TBI pathophysiology is characterised by acute necrotic or delayed apoptotic neuronal death, production of cytokines and chemokines, peripheral immune cell infiltration, and the initiation of microglia and astrocytes [66], which likely affects BGA through the immune pathway, destroying gut mucosal integrity. A study showed that gut NFκB increased after TBI, inducing an acute inflammatory response, including increased expression of intracellular adhesion molecule-1 (ICAM-1) and the generation of TNF-α, IL-6 and other cytokines [67]. Inflammatory cytokines lead to gut mucosal damage in rats via two mechanisms. First, these signals probably induce the downregulation of tight junction proteins that cause increased gut permeability. Second, the inflammatory response (especially by the generation of TNF-α) likely induces intestinal epithelial cell (IEC) apoptosis [68,69,70]. Additionally, research in humans indicated that TBI-induced alterations in the gut microbiome [71]. However, whether TBI affects BGA via m^6^A RNA modification is unclear, and the potential mechanisms are currently areas of investigation for the development of underlying interventions.

## 4. m^6^A RNA Modification and BGA

The accurate and controllable regulation of gene expression at the transcriptional and posttranscriptional levels is essential for the survival and function of eukaryotic cells. More than 100 RNA posttranscriptional modifications have been identified in eukaryotes. As one of the most common posttranscriptional RNA modifications in eukaryotes, m^6^A RNA modification can regulate gene expression [72,73]. Multiple lines of evidence from studies have shown that m^6^A modification influences almost all aspects of RNA metabolism, including RNA expression, splicing, nuclear output, translation, decay and RNA protein interactions [73,74,75,76]. Recent studies have revealed that m^6^A can delicately regulate a variety of spatial and temporal physiological processes, including gametogenesis, embryogenesis, sex determination, circadian rhythm, heat shock response, DNA damage response, cell fate determination, pluripotency, reprogramming and neuronal functions [75,77,78]. M^6^A plays a vital role in the development, differentiation, and regeneration of neurons [76,79,80]. It plays a crucial regulatory role in alterations in the gut microbiome [27], inflammatory gut diseases [81] and the development and progression of colon cancer [82]. Hui Wang et al. reported that the expression of m^6^A in the gut microbiome of patients with sepsis-associated encephalopathy (SAE) changed significantly [83]. In addition, our ongoing experiments revealed that the BGA dysfunction of YTHDF1-knockout mice following TBI showed significant differences compared to WT mice (Figure 4, preparing for publicity). Therefore, m^6^A exerts important effects on the bidirectional transmission of BGA.

### 4.1. m^6^A Related Genes and the Regulatory Mechanism of m^6^A Modification

The regulatory mechanism of m^6^A modification involves a dynamic and reversible process of methylation and demethylation [84]. M^6^A modification requires multiple regulatory proteins encoded by writing genes (writers), erasing genes (erasers) and reading genes (readers) [80] (Table 1). M^6^A is catalysed by a writer complex, consisting of core proteins methyltransferase-like 3 (METTL3), methyltransferase-like 14 (METTL14), and Wilms tumour 1-associated protein (WTAP) and several m^6^A-modified regulatory subunits. The first methyltransferase that binds to S-adenosylmethionine (SAM) in the methylase complex is METTL3 [85,86]. The binding of METTL3 and METTL14 is colocalised in nuclear spots to form a stable 1:1 heterodimer, which synergistically improves the methylation ability [87]. METT+L3/METTL14 heterodimers form in the cytoplasm and are recruited into the nucleus through nuclear localisation signals (NLSs) in METTL3 [88]. METTL3 can interact with the crimped part of WTAP via its lead helix regions and anchor it to RNA in the nucleus, which is rich in specific sites of premRNA processing factors [89]. WTAP interacts with zinc finger CCCH domain-containing protein 13 (ZC3H13) to keep the methyltransferase complex (MTC) in the nucleus, while ZC3H13 can interact with RNA binding motif protein 15 (RBM15) to recruit the MTC to U-rich sequences adjacent to the sequence RRACH (R = G/A; H = A/C/U) motif [90]. Another protein that interacts with WTAP is Virma, which provides sequence specificity and MTC recruitment to guide methylation in 3′UTR regions and near-stop codons [89]. Moreover, methyltransferase-like 16 (METTL16) [91], methyltransferase-like 5 (METTL5) and zinc finger CCHC domain containing 4 (ZCCHC4) act as linker proteins to guide MTC binding to its target mRNA [84]. To date, the last protein in MTC is Hakai, which plays a role in the sex-determining pathway and mediates sex-lethal splicing [18]. In addition, what is needed to maintain the integrity of m6A-METTL-Associated Complex (MACOM) is the ubiquitin domain of Hakai but not its activity (Figure 5). Hakai is essential to maintaining the functionality of m^6^A writers by maintaining the stability of MACOM components [92].

The demethylases (erasers) that reverse m^6^A methylation are FTO and ALKBH5. The FTO demethylation process requires ferrous ions and α-ketoglutarate to demethylate m^6^A via a sophisticated intermediate reaction [84]. First, FTO oxidises N6-methyladenosine to form N6-hydroxymethyladenosine (hm^6^A), which then catalyses the formation of N6-formyladenosine (f^6^A). Finally, f^6^A generates adenosine (A) (Figure 6), and the demethylation process is completed [120]. Intriguingly, a recent finding showed that FTO preferentially demethylated N6,2′-O-dimethyladenosine (m^6^A_m_) rather than m^6^A; by altering FTO levels, m^6^A_m_ at the mRNA cap weakened the unwrapping effect of mRNA by decapping RNA2 (DCP2), and increasing m^6^A_m_ levels through FTO knockout also improved the stability of mRNAs [121]. Additionally, FTO regulates reversible m^6^A_m_ RNA methylation in small nuclear RNA (SnRNA) biogenesis [108,122]. These studies indicate that internal mRNA m^6^A sites may not be related to FTO substrates. However, Jiang Bo et al. found that FTO could effectively demethylate both m^6^A and cap m^6^A_m_ from purified polyadenylated RNA with different substrate preferences in the nucleus and cytoplasm [107]. ALKBH5 is a kind of m^6^A demethylase that exhibits no activity against m^6^A_m_ but does act strongly against m^6^A [121]. ALKBH5 mediates the stability of target mRNA [123] and reduces the output of mRNA to the cytoplasm [124]. These findings have begun to shed new light on multiple intricate m^6^A demethylation mechanisms.

M^6^A reader proteins modulate m^6^A modification biological functions to recognise m^6^A sites [125]. The proteins involving YTH domains are best characterised as m^6^A “readers” [126], and there are five YTH domain-involving proteins (YTHDF1-3, YTHDC1, and YTHDC2) in mammals [127]. YTHDF1 promotes mRNA translation in the classical model, YTHDF2 facilitates mRNA degradation, and YTHDF3 cooperates with YTHDF1 or YTHDF2 to promote either mRNA translation or degradation [19,126]. However, recent findings confirmed that the m^6^A sites binding with YTHDF1, YTHDF2, or YTHDF3 are highly similar, and that all three analogues act in concert to regulate the degradation of m^6^A-tagged mRNA [93,112]. Using dCasRx-M3 and dCasRx-A5 to regulate the methylation level of targeted sites at YTHDF paralogues, Zhen Xia et al. found that the m^6^A sites binding with YTHDF1, YTHDF2, and YTHDF3 all led to m^6^A-mediated mRNA degradation [93], which supported recent models [93,112] rather than the classical model [19,126]. YTHDC1 is the only YTH family protein located in the nucleus [128]. Previous studies have shown that YTHDC1 plays a crucial role in mediating mRNA splicing by recruiting serine/arginine-rich splicing Factor 3 (SRSF3), blocking serine/arginine-rich splicing Factor 10 (SRSF10) [110], and retaining m^6^A-modified exons during splicing [129]. Moreover, YTHDC1 also regulates the nuclear and cytoplasmic transport of methylated mRNAs [110]. Recently, however, Zhijian Qian’s team found that YTHDC1 KO did not influence the splicing of minichromosome maintenance deficient 4 (MCM4) in leukemic cells and did not affect the nuclear output and translation efficiency of MCM4 and MCM5 in acute myeloid leukaemia (AML) cells; in contrast, YTHDC1 mediated the expression of MCM2, MCM4 and MCM5 transcripts by controlling the stability of these transcripts in AML cells [128]. YTHDC2, the last member of the YTH protein family, which is chiefly enriched in the perinuclear region, can bind to ribosomes and function in small ribosomal subunits, decrease mRNA abundance and enhance hypoxia-inducible factor 1α (HIF-1α) mRNA translation efficiency by its helicase [114,130]. YTHDC2 harbours a unique domain that differs from other YTH family proteins. The YTHDC2 RNA binding domain, characterised by the R3H domain, binds YTHDC2 to intracellular RNA and plays an auxiliary role in the YTH domain [113,114]. In addition to YTH proteins, heterologous nuclear ribonucleoproteins A2B1 (HnRNPA2B1) and C (HNRNPC), two nuclear RNA-binding proteins, play a vital role in m^6^A-dependent nuclear RNA processing events [131]. HnRNPA2B1 directly binds to m^6^A sites of RNA and interacts with the microRNA microprocessor complex protein DiGeorge syndrome critical region gene 8 (Dgcr8) to promote METTL3-mediated microRNA processing [84,115]. HNRNPC interacts with m^6^A-modified mRNA to mediate structural transformation [117]. However, unlike HNRNPC, HNRNPG binds to a purine-rich motif of m^6^A-methylated RNAs, including the m^6^A site, via its low-complexity region at the C-terminus, which is self-assembled into large particles in vitro. Due to the overlap of the HNRNPG binding motif and the m^6^A site, HNRNPG binding may compete with the binding of direct m^6^A readers such as YTH domain proteins [116]. EIF3 is a large multiprotein complex composed of 13 subunits (a-m) [132]. By binding to 5′UTR m^6^A residues, EIF3 can directly recruit the 43S preinitiation complex to mRNA 5′UTRs to stimulate translation initiation [118]. Insulin-like growth Factor 2 mRNA binding protein 1, 2, and 3 (IGF2BP1, IGF2BP2 and IGF2BP3) also promotes target mRNAs’ stability and translation [119]. Intriguingly, Ras GTPase-activating protein-binding protein 1 (G3BP1) and G3BP2, two anti-readers of m^6^A, have been confirmed to date to act on the assembly of stress granules and are essential for embryonic development [133] but are expelled by the existence of m^6^A [134,135].

### 4.2. m^6^A Modification and Brain

M^6^A modification is abundant in the brain, normalising the nervous system function by regulating mRNA metabolism. When the expression of key enzymes and/or binding proteins involved in m^6^A modification is abnormal, the level of m^6^A modification may fluctuate, leading to molecular dysfunctions such as mRNA metabolism disorder and transcription and translation disorders that cause physiological dysfunction of the nervous system [136,137,138]. Given its diversity and reversibility, m^6^A modification harbours more complicated physiological mechanisms in regulating CNS functions. For writers, METTL3 can influence the mRNA half-life of Dapk1, Fadd, and Ngfr. Indeed, the deletion of METTL3 can ectopically modify m^6^A, leading to cerebellar hypoplasia, long-term memory lesion, and effects on the self-renewal and differentiation of stem cells [139]. METTL14 can recognise and promote Pten protein translation, while METTL14 knockout can affect axonal regenerative ability [140]. ZC3h13 keeps the Zc3h13-WTAP-Virilizer-Hakai complex in the nucleus and enhances the level of mRNA m^6^A modification in a mouse embryonic stem cell (mESC). ZC3h13 knockout may cause the complex to be transferred from the nucleus to the cytoplasm, reducing the level of m^6^A, which destroys the self-renewal of mESCs and induces mESC differentiation [101]. For erasers, FTO can repress synaptophysin (Syp) gene mRNA transcription in synaptic ganglia and reduce Spy protein levels; downregulating the expression of FTO can also upregulate Syp protein levels and improve memory [141]. ALKBH5 decreases the m^6^A modification of C-X-C motif chemokine ligand 2 (CXCL2) and IFN-CD4 mRNA. It increases transcriptional stability and protein expression, which results in the enhanced response of CD4+ T cells and more neutrophils infiltrating the CNS in the process of neuroinflammation [142]. For readers, YTHDF1 recognises and binds the mRNAs of m^6^A-modified GRIN1, GRIN2A, GRIA1, CAMK2A, and CAMK2B genes to promote translation. Indeed, YTHDF1 deletion causes ectopic translation of related proteins, resulting in memory loss [79]. YTHDF2 reduces mRNA stability and facilitates its degradation by recruiting the CCR4-nondead enolase complex. Deleting YTHDF2 in the embryonic neocortex seriously influences the self-renewal of neural stem/progenitor cells (NSPCs). The spatiotemporal generation of neurons and other types of cells leads to death in the later stage of embryonic development [143]. YTHDF3 can specifically recognise m^6^A-modified Apc gene mRNA in the cytoplasm and regulate the translation process with YTHDF1. Deleting YTHDF1 and YTHDF3 can cause abnormal translation of Apc protein, which can induce ectopic neuronal development and attenuate synaptic transmission ability [144]. YTHDC1 plays a role in premRNA splicing and mediates the nuclear and cytoplasmic transport of methylated mRNAs, thus facilitating neuronal survival and reducing ischaemic stroke by destroying PTEN mRNA and promoting Akt phosphorylation [127]. YTHDC1 knockout downregulates the expression of the antiapoptotic protein Bcl2. It upregulates the expression of the proapoptotic protein cleavage caspase3, while the overexpression of YTHDC1 confirms its protective effect on neuronal death induced by oxygen-glucose deprivation (OGD) [127]. YTHDC2 can bind to ribosomes and act on small ribosomal subunits, decrease mRNA abundance and enhance HIF-1α mRNA translation efficiency by its helicase [114]. YTHDC2 can act as an independent gene as a prognostic marker of head and neck squamous cell carcinoma (HNSCC) [130]. Alteration of m^6^A modification exerts significant effects on brain tissue development, learning and memory ability, stem cell renewal and differentiation, synaptic regeneration, and other biological functions (Figure 7).

Therefore, the abnormal expression of m^6^A-related proteins may lead to the development of various neurological diseases, such as PD [145,146,147], AD [148,149,150,151,152], MS [153], tumours [154,155,156], epilepsy [157], and neuropsychiatric disorders [158]. TBI, as one of the major diseases of CNS, often causes memory problems in humans [159]. Several CNS traumas, both in vitro and in vivo experimental systems [147,160,161,162,163,164], have demonstrated that m^6^A modification also plays important roles in the aetiology and pathogenesis of brain injuries. Diao et al. reported that m^6^A levels in the total RNA of primary cultured rat hippocampal neurons increased markedly after hypoxia/reoxygenation (H/R) injury [161]. The expression of total RNA m^6^A modification also upregulated significantly in the cerebral cortex of rats and mice treated with middle cerebral arteryocclusion/reperfusion (MCAO/R) [160,162]. However, a decreased level of the total RNA m^6^A was observed with the elongation of reperfusion time in PC12 cells with oxygen-glucose deprivation/reoxygenation (OGD/R) treatment [162]. The total RNA m^6^A levels of the hippocampus of brain-injured mice decreased significantly at 6 h after injury compared with that in the sham operation group in a controlled cortical impact (CCI) mouse model [164], while the change in m^6^A methylation reduction only occurred in the striatum in a rat nerve injury model treated with 6-hydroxydop amine (6-OHDA) [147]. These studies revealed that the level of total RNA m^6^A modification differs between brain injury models, indicating that the cell type and brain region might affect the m^6^A modification level [165]. More importantly, the m^6^A-related enzymes are differentially expressed in CNS injuries and play significant roles in neuronal development, cognitive disorders, cell death, and apoptosis, suggesting that they may serve as a novel biomarker and therapeutic target for CNS injuries [166].

### 4.3. m^6^A Modification and Gut Microbiome

The gut microbiome influences host physiology, including host metabolism, gut morphology, immune system development, and behaviour [167,168]. Gut microbial metabolites and fermentation products, such as short-chain fatty acids (SCFAs), tryptophan metabolites, sphingolipids, and polyamines, have been known to partially regulate the effect of the gut microbiome on the host by modulating transcription and epigenetic modifications [169,170,171,172]. Sabrina Jabs et al. found that m^6^A modification spectra in both the mouse ceca and liver were affected by the presence of the gut microbiome by MERIP-Seq [27]. Xiaoyun Wang et al. carried out mass spectrometry and m^6^A sequencing on GF and SPF mice brain, gut, and liver tissues. The results revealed that the gut microbiome harbours a significant effect on the m^6^A mRNA modification of the host brain, gut, and liver, but especially in the brain [173]. In addition, dietary intake of non-starch polysaccharides (NSPs) may alter m^6^A RNA methylation to improve tumours by affecting the supply of methyl donors produced by the gut microbiome [174]. Multiple studies have shown that m^6^A modification is the key interaction mechanism between the gut microbiome and host, and m^6^A modification may also selectively modulate microbiome function by affecting gut inflammation. A recent study indicated that METTL3-knockout mice developed chronic gut inflammation at 3 months after birth, which resulted in gut microbiome disorders [175]. METTL14 deficiency interferes with the induction of immature T cells into inducted regulatory T cells (Tregs), which results in spontaneous colitis in mice [81]. Benyu Liu et al. found that CircZbtb20 promotes the interaction between Alkbh5 and Nr4a1 mRNA, resulting in the ablation of Nr4a1 mRNA m^6^A modification to enhance the stability of Nr4a1 mRNA. Nr4a1 initiates Notch2 signalling activation, which helps to maintain the dynamic balance of innate lymphoid cells (ILC3). The absence of Alkbh5 or Nr4a1 disrupts ILC3 homeostasis and causes serious gut diseases, such as inflammation [176]. YTHDF1 guides the translation of tumour necrosis factor receptor-associated factor 6 (TRAF6) mRNA, which encodes TRAF6, thus modulating the immune response by m^6^A modification near the transcript stop codon. Moreover, YTHDF1 recognises the target TRAF6 transcript to modulate the gut immune response to bacterial infection by the unique interaction mechanism between the P/Q/N-rich domain and host factor DDX60 death domain [177]. Ane Olazagoitia-Garmendia et al. demonstrated that a gluten-induced increase in YTHDF1-mediated XPO1 activates the NFκB pathway and then induces increased expression of IL8 in gut cells, resulting in the development and aggravation of celiac disease [178]. Dian Xu et al. used The Cancer Genome Atlas (TCGA) cohort data to analyse the expression patterns of m^6^A RNA methylation regulators in colonic adenocarcinoma (COAD) and their relationship with clinical features. The results suggested that YTHDF1 exhibited the highest diagnostic value for COAD, and survival analysis confirmed that highly expressed levels of YTHDF2 and YTHDF3 indicated a favourable prognosis. Furthermore, the expression levels of YTHDF3 were independent prognostic factors of 5-year total survival in COAD patients [82]. Tanabe et al. reported that YTHDC2 was upregulated in colon cancer and positively associated with tumour stages [179]. These findings confirm that host m^6^A modification also provides feedback to gut bacteria to sustain gut homeostasis.

## 5. The Role of m^6^A RNA Modification in TBI-Mediated BGA

### 5.1. m^6^A Modification and TBI

After TBI, excitatory amino acids such as glutamate release into the synapse which then overstimulates the N-methyl-D-aspartate (NMDA) receptor, which leads to Ca^2+^ overload [180,181]. Ca^2+^ overload and excitotoxicity induce reactive oxygen species (ROS) release [180], and excessive ROS causes oxidative stress and neuronal cell death [182,183]. Oxidative stress induces excessive free radicals, resulting in neuronal death, stimulating inflammation [184]. Oxidative stress occurs at the beginning and accompanies the whole process of TBI and is one of the significant mechanisms. A large number of studies have confirmed that m^6^A plays an important role in oxidative stress [151,185,186,187,188]. Anders et al. first found that the m^6^A signal accumulates in reaction to oxidative stress in stress granules and that YTHDF3 promotes target mRNA translocation into stress granules [189]. Ye Fu et al. observed that YTHDF1/3-knockout largely abolishes stress granule formation and mRNA localisation to stress granules. Reintroducing YTHDF proteins into knockout cells restores the formation of stress granules [190]. The regulation of phase separation by m^6^A weakens the stability and translation of mRNA, influencing cellular functions and biological processes [84]. Wang, J. et al. identified that METTL3 may have a protective effect against oxidative stress [191].

TBI-induced oxidative stress can alter genes expression and signal pathways, such as nucleotide-binding oligomerization domain-like receptor pyrin domain-containing-3 (NLRP3) [192], caveolin-1 [193], and p75-neurotrophin receptor (p75NTR) [194], and nuclear factor erythroid 2-related factor 2 (Nrf2) signalling [192,195], etc. These alterations may be correlated with posttranscriptional regulation, especially m^6^A. Considering the indispensable function of m^6^A modification in mediating protein translation and participating in various bioprocesses, it has been confirmed that m^6^A modification has also recently contributed to the aetiology and pathogenesis of TBI [163,164,165]. Yiqin Wang et al. investigated novel epigenetic alterations in TBI via genome-wide screening of m^6^A marker transcripts in the mouse hippocampus after TBI [164]. The results revealed that the expression of METTL3 and the total level of RNA m^6^A in the mouse hippocampus were distinctly decreased after TBI. The immunohistochemical results showed that METTL3 was mainly expressed in neurons, suggesting that cognitive dysfunction may be correlated with the downregulation of METTL3. Jiangtao Yu et al. found that the expression of METTL14 and FTO in the rat cortex was decreased after TBI and that there was a significant change in the m^6^A methylation levels of 1580 mRNAs. The study also revealed that FTO harboured a significant role in maintaining neurological functions in TBI rat that would be an intervention target for mitigating impairments caused by TBI [163]. However, it is not clear whether the change in TBI-induced m^6^A methylation is associated with BGA dysfunction.

### 5.2. The Role of m^6^A in TBI-Mediated BGA

It is known that TBI can enhance the expression of NLRP3 [192,196] and caveolin-1 [193], and impede Nrf2 signalling [192,197], and these changes are also displayed in gut diseases [198,199,200]. Among them, NLRP3 and Nrf2 are the potential mediators in BGA dysfunction [198,201,202]. Moreover, there is clear evidence that m^6^A mediates gene expression and the signal pathway [191,203,204]. Hence, it is reasonable to speculate that m^6^A may take part in TBI-induced BGA dysfunction via regulating the expression of NLRP3, caveolin-1, and Nrf2 signalling (Figure 8).

#### 5.2.1. METTL14/TINCR/NLRP3 Axis

METTL14-mediated m^6^A modification induces inhibition of non-protein coding RNA (TINCR), and thus represses NLRP3 mRNA stability, preventing the development of pyroptosis [203]. In contrast, deletion of METTL14 causes upregulated NLRP3, cleaved caspase-1, and gasdermin D (GSDMD)-N [203]. NLRP3 inflammasome is a multiprotein complex and regulates inflammatory responses by activating caspase-1 for processing premature IL-1β and IL-18 [205]. NLRP3 inflammasome may be a potential target for treatments and biomarkers to improve TBI prognosis [206,207]. Multiple animal studies indicate an upregulation of the NLRP3 inflammasome at both gene and protein levels after a moderate TBI induced via CCI [208,209,210]. Furthermore, there is also clinical evidence of NLRP3 inflammasome potentiate in moderate and severe TBI [211,212]. Moreover, NLRP3 inflammasome is at the crossroads of BGA communications [213]. Taken together, TBI-induced METLL14 downregulation may lead to neuroinflammation through the METTL14/TINCR/NLRP3 axis, further causing BGA dysfunction via the immune pathway.

#### 5.2.2. METLL3/miR-873-5p/Keap1/Nrf2 Signalling Pathway

METTL3-mediated m^6^A modification enhances DGCR8 recognition of pri-miR-873, promoting the generation of mature miR-873-5p. MiR-873-5p inhibits Kelch-like ECH-associated protein 1 (Keap1) and then activates the Nrf2//heme oxygenase-1 (HO-1) pathway [191]. TBI can impair autophagic flux and impede Nrf2 signalling, the major mediator in antioxidant response, thus leading to excessive oxidative stress [197]. Furthermore, the expression of NLRP3 inflammasome proteins can be induced by oxidative stress through increasing the expression of key proteins of the Nrf2/HO-1 pathway following TBI [192]. Additionally, Nrf2 activators have an influence on mitochondrial quality control and are correlated with a positive change in BGA [202]. Hence, TBI-induced METLL3 downregulation may induce oxidative stress and neuroinflammation via the METLL3/miR-873-5p/Keap1/Nrf2 signalling pathway, consequently leading to BGA dysfunction.

#### 5.2.3. FTO/Caveolin-1/MMP2/9 Pathway

FTO directly targets the caveolin-1 mRNA and facilitates its degradation [204]. Thus, attenuated FTO in the rat cortex enhances caveolin-1 expression after TBI. Potentiated caveolin-1 expression and phosphorylation are associated with the decreased expression of occludin and claudin-5 [214,215]. Caveolin-1 activates matrix metalloproteinases (MMPs), especially MMP2 and MMP9, promoting BBB opening after injury [216,217,218]. MMPs are a group of Zn^2+^ dependent endonucleases that can cleave type IV collagen in the extracellular matrix of the BBB, inducing structure damage and leakage [219]. MMP2 and MMP9 activated by zinc accumulation in microvessels, especially, mainly cause the loss of TJs complex to destroy the BBB [220]. In addition, clinical datasets showed that disease-active patients displayed enhanced caveolin-1 level compared to normal controls or disease-inactive inflammatory gut disease groups [199]. Altogether, TBI-induced FTO downregulation may cause the loss of TJs complex via the FTO/caveolin-1/MMP2/9 pathway, ultimately leading to the destruction of the BBB and the gut barrier.

After TBI, a series of pathological reactions produced by brain tissue affect the integrity of the gut mucosa and the constituents and colonization of the gut microbiome via BGA signal transduction. Given that m^6^A plays an important role in regulating these pathological processes, it hints that m^6^A may take part in pathological processes of BGA after TBI via the above pathways. However, these studies are limited, and there may be other regulatory pathways that have not been found to date. The specific mechanisms behind m^6^A in BGA (especially after TBI) remain elusive. The important molecular information transmission processes m^6^A involved in post-TBI BGA require in-depth study.

## 6. Conclusions and Perspectives

M^6^A methylation plays an important role in the homeostasis regulation of tissues and organs, especially when injury occurs, such as TBI, oxidative stress, ischemic perfusion, etc. The role of m^6^A in the brain and gut and the microbiome has been demonstrated. Therefore, it is reasonable to speculate that m^6^A may take part in TBI-induced BGA dysfunction.

Although BGA posttranscriptional regulation studies have developed rapidly, many questions are still existing. First, although it has been confirmed that m^6^A plays a significant role in BGA, the specific role of each enzyme and protein in BGA and its detailed mechanism are not clear. Second, there is currently no report on the treatment of BGA dysfunction after TBI with specific agonists or inhibitors of m^6^A-related enzymes and proteins. It was reported that vitagenes play a key role in the pro-survival pathways involved in the programmed cell life, conferring protection against oxidative stress [221,222,223]. Vitagenes can promote the production of molecules (bilirubin, etc.) harbouring antioxidant and antiapoptotic activities [224]. To date, mountains of reports have shown that bilirubin is endowed with strong antioxidant activity, alleviating cellular damage induced by reactive oxygen species in vitro and in vivo models [225,226,227]. The process of vitagenes protect against oxidative stress causing cellular damage may be regulated by the Keap1/Nrf2/antioxidant response element (ARE) pathway [224]. The neuroprotective effects of the upregulation of the Keap1/Nrf2/ARE pathway have been confirmed in several in vitro and in vivo experimental systems [228,229]. Given that the Nrf2 signalling pathway plays an important role in neuroinflammation caused by oxidative stress after TBI [230,231], the upregulation of the Keap1/Nrf2/ARE pathway may block the occurrence of neuroinflammation caused by oxidative stress after TBI, thus producing neuroprotective effects. The polyphenolic compounds can attenuate oxidative stress-induced traumatic brain injury and reduce secondary injury via the Nrf2 signalling pathway [192,230,231]. Indeed, substantial lines of evidence have demonstrated that m^6^A plays an important role in oxidative stress by modulating the Nrf2 signalling pathway [187,191,232,233]. Meanwhile, as mentioned above, m^6^A may regulate TBI-induced BGA dysfunction via the METLL3/miR-873-5p/Keap1/Nrf2 signalling pathway. A member of our group, Xiang Zhong, reported that phytochemicals can interfere with the expression of m^6^A RNA methylation in preclinical experiments [234,235,236,237,238]. Thus, the polyphenolic compounds may act as potential drugs for the prevention and treatment of BGA dysfunction after TBI through regulating the m^6^A-related enzymes and proteins. Though there is still limited clinical evidence for these natural Nrf2 activators, we foresee that the combinational use of phytochemicals such as Nrf2 activators with gene and stem cell therapy such as agonists or inhibitors of specific RNA m^6^A-related enzymes and proteins will be a promising therapeutic strategy for TBI and TBI-induced BGA dysfunction in the future.

## Figures and Tables

**Figure 1 antioxidants-11-01521-f001:**
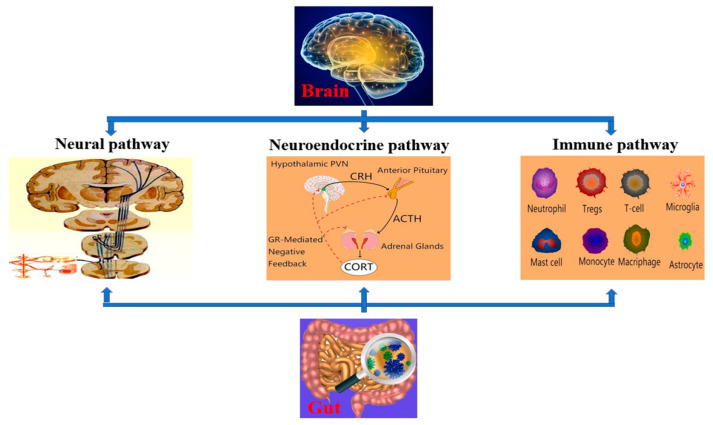
Schematic representation of the 3 pathways (neural, neuroendocrine and immune) that constitute the brain–gut axis. (1). Neural pathway: the brain regulates gut function mainly through ANS, while the gut microbiome exerts a feedback effect on the brain. (2). Neuroendocrine pathway: stimulation of the brain is transmitted to the gut via the HPA axis, while the gut microbiome influences the brain by affecting the production of immune mediators to activate the HPA axis. (3). Immune pathway: the brain causes gut dysfunction via releasing proinflammatory cytokines to recruit other immune cells, while the gut microbiome disrupts BBB integrity by downregulating TJs expression.

**Figure 2 antioxidants-11-01521-f002:**
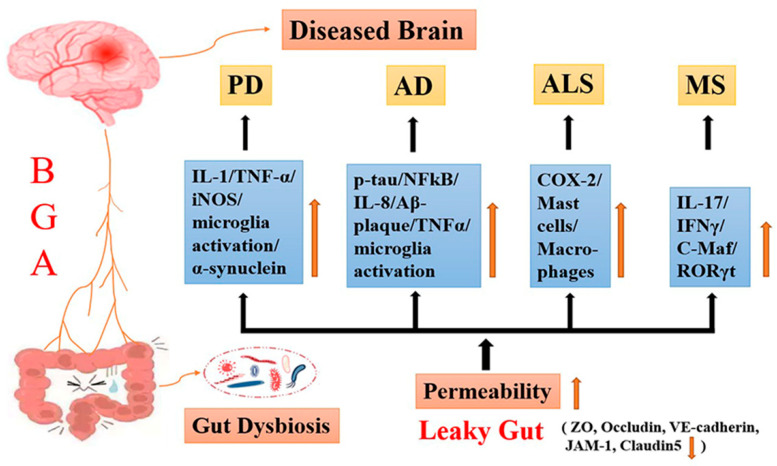
Diagrammatic representation of the gut dysbiosis leads to the development of neurodegenerative diseases and their potential mechanisms through activation of inflammatory pathways.

**Figure 3 antioxidants-11-01521-f003:**
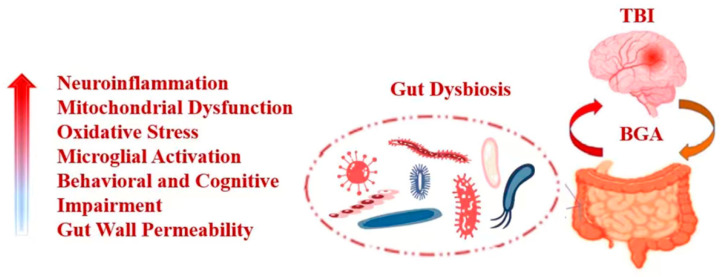
TBI caused dysbiosis through the BGA and its negative feedback mechanism.

**Figure 4 antioxidants-11-01521-f004:**
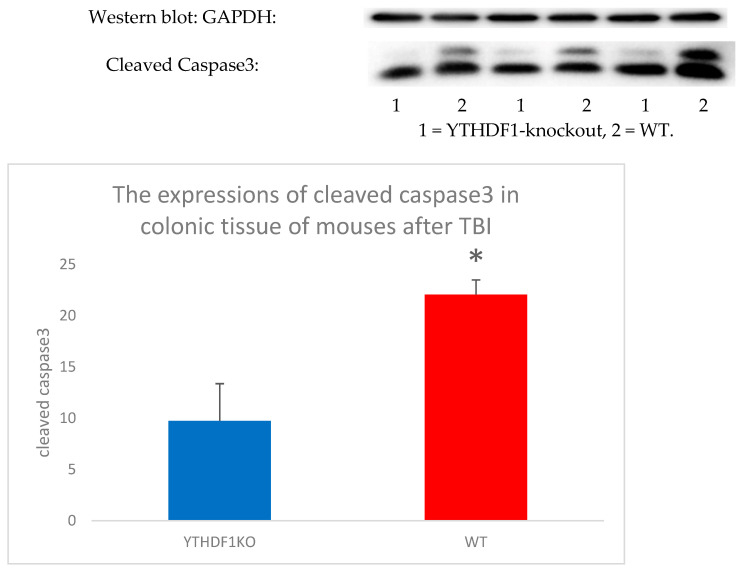
The expressions of cleaved caspase3 in colonic tissue of YTHDF1-knockout mice is significantly downregulated after TBI compared to WT mouses. * *p* < 0.05.

**Figure 5 antioxidants-11-01521-f005:**
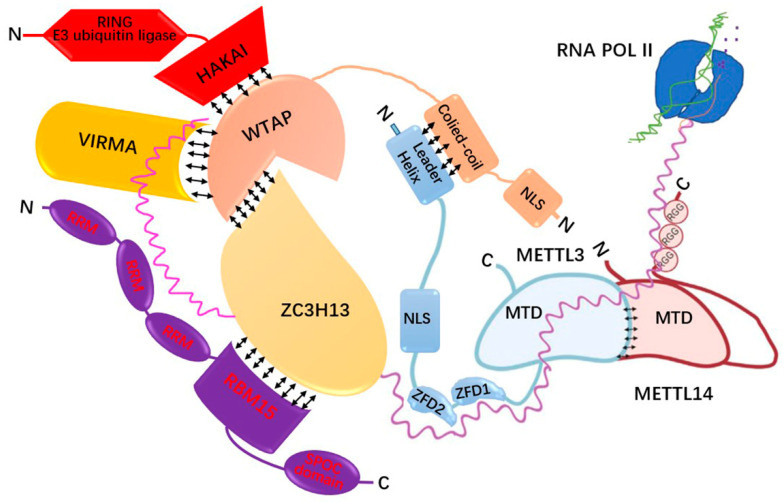
Schematic diagram of the methyltransferase complex: The components and their interactions.

**Figure 6 antioxidants-11-01521-f006:**
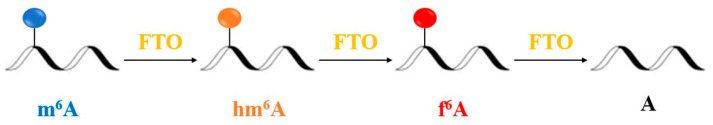
Schematic diagram of the process of FTO demethylates m^6^A.

**Figure 7 antioxidants-11-01521-f007:**
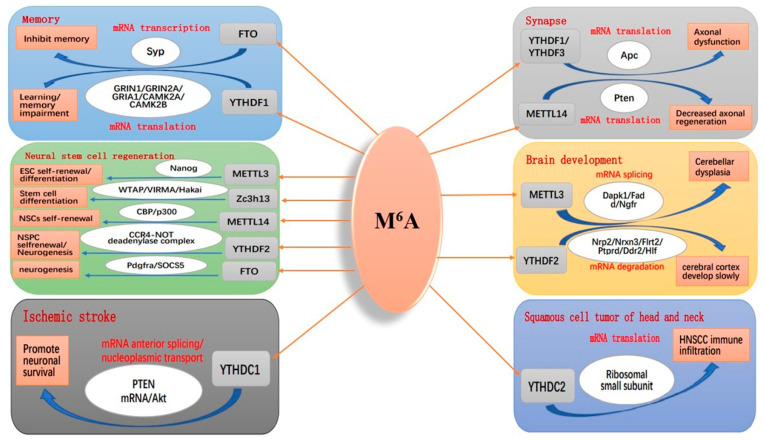
The role of m^6^A modification in the CNS.

**Figure 8 antioxidants-11-01521-f008:**
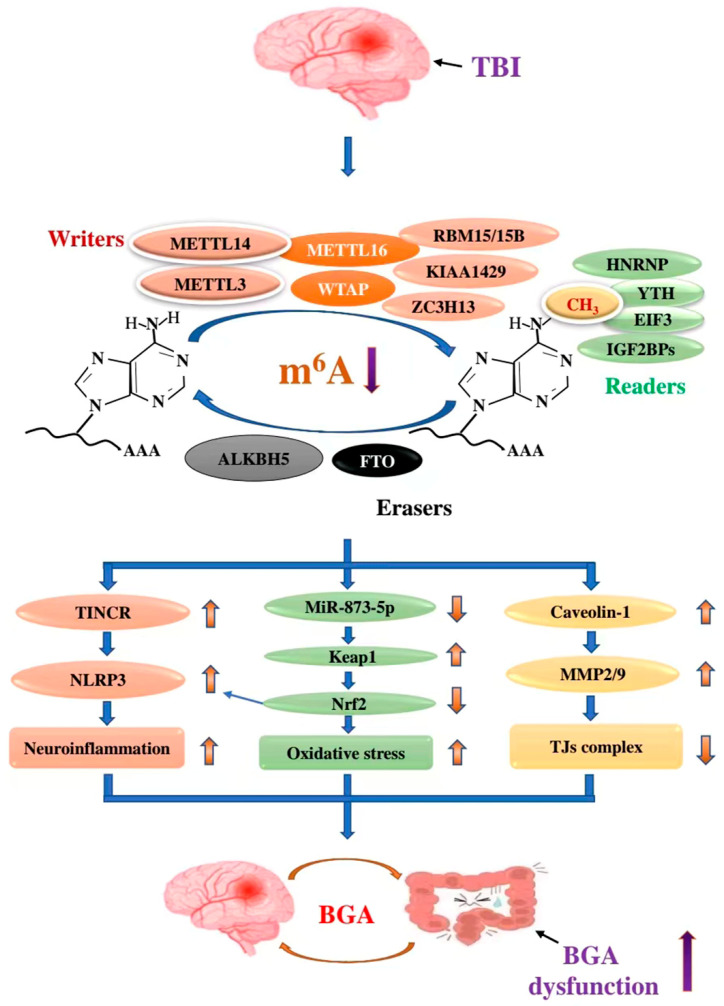
The possible pathways of m^6^A regulating the pathological process of BGA after TBI: TBI causes the downregulation of m^6^A, which is involved in TBI-induced BGA disfunction via METTL14/TINCR/NLRP3, METLL3/miR-873-5p/Keap1/Nrf2 signalling, and FTO/Caveolin-1/MMP2/9 pathways.

**Table 1 antioxidants-11-01521-t001:** M^6^A related genes and their basic functions.

Types	Regulators	Functions	References
m^6^A writers	METTL3	Catalyses the transfer of the methyl in single-stranded RNA (ssRNA) sequence motif DRACH (D = A, G or U; R = A or G; H = A, C or U) from S-Adenosyl methionine (SAM) to adenine.	[87,93]
	METTL14	Recognizes the RNA substrate that activates METTL3 and offers RNA binding sites as scaffolds to form a stable heterodimer with METTLL3.	[84,87,93]
	WTAP	The first one binds to METTL3-METTL14 heterodimer and recruits it to target RNA. The three proteins form together a conservative complex located in nuclear spot.	[84,94,95]
	METTL16	Responsible for m^6^A modification of lncRNAs, U6 snRNA, and introns of pre-mRNAs.	[91,96,97,98]
	RBM15	Binds m^6^A complex and recruits it to a special RNA site.	[84,99]
	VIRMA	Recruits m^6^A complex to a special RNA site and interacts with polyadenosine cleavage factors CPSF5 and CPSF6.	[89,100]
	ZC3H13	Bridges WTAP to mRNA binding factor Nito.	[84,101]
	METTL5	Responsible for m^6^A modification of 18s rRNA.	[96,102,103]
	ZCCHC4	Responsible for m^6^A modification of 28s rRNA.	[96,104,105,106]
	HAKAI	Exerts effects on gender determination and mediates lethal splicing, maintains the functions of m^6^A writers by ensuring the stability of MACOM components via the Hakai ubiquitin domain.	[92]
m^6^A erasers	FTO	Demethylates m^6^A, also harbours activity towards m^6^Am and m^1^A.	[96,107,108,109]
	ALKBH5	Mainly demethylates m^6^A.	[96,110,111]
m^6^A readers	YTHDF1/2/3	Highly similar to m^6^A sites bound by YTHDF1, YTHDF2 or YTHDF3, and these three analogues together exerts effects on mediating mRNA degradation.	[93,112]
	YTHDC1	Promotes alternative splicing and RNA output.	[96,110,111]
	YTHDC2	Boosts target RNA translation and reduces its abundance.	[96,113,114]
	HNRNPA2B1	Mediates mRNA splicing and major microRNA processing.	[84,115]
	HNRNPC/ hn-RNPG	Regulates mRNA structure and alternative splicing.	[96,116,117]
	EIF3	Facilitates mRNA translation.	[84,118]
	IGF2BP1/2/3	Enhances mRNA stability, storage capacity and translation.	[96,119]

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
