# Peer review of "The Potential Role of m6A in the Regulation of TBI-Induced BGA Dysfunction"

_antioxidants, 2022, doi:10.3390/antiox11081521_

Round 1

Reviewer 1 Report

Authors tried to address the question about the role of RNA m6A modification in BGA dysfunction following TBI. The relationship between TBI and BGA dysfunction, and the potential role of m6A in BGA has been well summarized. However, so far there are not enough published data or directly evidences ( only 2 papers related to m6A and TBI, while no any report directly exploring the function of m6A in BGA post-trauma) to support the key idea: m6A regulates BGA dysfunction following TBI. This idea is just an assumption. Authors need more coming out data to support the hypothesis or draw a conclusion. 

Reviewer 2 Report

The manuscript focuses on the role of M6a RNA in gut-brain axis dysfunction induced following traumatic brain injury.

The text is tiring to read as a whole. But it is very summary in the first part (up to about line 150).

Figures 1 and 2 are of low quality and stretched horizontally.

Some additional figures might help (for example, a graph between lines 156-196.

The paragraph (line 286-294) is very vague.

I suggest inserting a section of Abbreviations.

Reviewer 4 Report

An interesting and quite new approach.

I really enjoy it reading the manuscript.

It is of interest for the readers, for sure.

Before acceptance, some minor issues:

1. There seems to be some technical deficiencies with numbering the references. There are 2 numbers there. This is very misleading to the reader and for the scientific accuracy of the info provided in [] in the text. Please repair it

2. Some 2022 papers on TBI could be added in the references, since this is a growing area and needs the latest updates. Even in MDPI I found several papers in 2022, not to mention MEDLINE etc

3. I think you should modify the title of the last section in “Conclusions and Perspectives” not prospect

4. Although you mention the oxidative stress quite early in the conclusion, you only mentioned it as a possible mechanistical pathway in a single paragraph in the next. Please develop more on that matter.

Reviewer 5 Report

This is a decent review devoted to a hot topic in the field. The text and the figures are informative. My concern is the inability to find any previous experimental or theoretical contribution of any co-author in the field of the review.

Round 2

Reviewer 1 Report

The authors' response partially addressed the question. Due to lacking of directly published data to support the hypothesis, the title would be reshaped. For example,  " The potential role of M6A in regulation of TBI induced BGA dysfunction". Meanwhile, merging the data listed in response into manuscript will strongly support the hypothesis. 

Reviewer 2 Report

The authors did not completely satisfy the suggested requests, as the first part of the manuscript remained summary, and a section of Abbreviations was not included.

Reviewer 3 Report

I found the manuscript has been sufficiently update in conceptualization and arguments for publication in ARS journal. Please, just check out the English style that needs further corrections.
